# Anti-Inflammatory Effect of Simonsinol on Lipopolysaccharide Stimulated RAW264.7 Cells through Inactivation of NF-κB Signaling Pathway

**DOI:** 10.3390/molecules25163573

**Published:** 2020-08-06

**Authors:** Lian-Chun Li, Zheng-Hong Pan, De-Sheng Ning, Yu-Xia Fu

**Affiliations:** Guangxi Key Laboratory of Functional Phytochemicals Research and Utilization, Guangxi Institute of Botany, Chinese Academy of Sciences, Guilin 541006, China; llc@gxib.cn (L.-C.L.); ndsh@gxib.cn (D.-S.N.); fuyusha88@163.com (Y.-X.F.)

**Keywords:** simonsinol, anti-inflammatory, *Illicium simonsii*, sesqui-neolignan, NF-κB

## Abstract

Simonsinol is a natural sesqui-neolignan firstly isolated from the bark of *Illicium simonsii*. In this study, the anti-inflammatory activity of simonsinol was investigated with a lipopolysaccharide (LPS)-stimulated murine macrophages RAW264.7 cells model. The results demonstrated that simonsinol could antagonize the effect of LPS on morphological changes of RAW264.7 cells, and decrease the production of nitric oxide (NO), tumor necrosis factor α (TNF-α), and interleukin 6 (IL-6) in LPS-stimulated RAW264.7 cells, as determined by Griess assay and enzyme-linked immunosorbent assay (ELISA). Furthermore, simonsinol could downregulate transcription of inducible nitric oxide synthase (iNOS), TNF-α, and IL-6 as measured by reverse transcription polymerase chain reaction (RT-PCR), and inhibit phosphorylation of the alpha inhibitor of NF-κB (IκBα) as assayed by Western blot. In conclusion, these data demonstrate that simonsinol could inhibit inflammation response in LPS-stimulated RAW264.7 cells through the inactivation of the nuclear transcription factor kappa-B (NF-κB) signaling pathway.

## 1. Introduction

Inflammation is a complex biological defensive mechanism of vascular organisms. It is activated when the host is exposed to injury factors, and subsequently, triggers off a series of biological responses to repair tissue, which are meditated mainly by inflammatory meditators [1,2]. However, dysfunction of inflammation has been demonstrated that is associated with many human illnesses, including cancer, diabetes, and rheumatoid arthritis [3,4,5]. Therefore, anti-inflammatory drugs are needed to regulate abnormal inflammation response. Natural products, especially those that exist in medical plants, are a promising and important source of anti-inflammatory drugs, and many anti-inflammatory drugs already on the market are derived from natural products in medical plants [6,7].

Discovering novel and bioactive natural products from medical plants is one of our team’s research interests. Recently, our team isolated four sesqui-neolignans from the bark of *Illicium difengpi*, including two new compounds, difengpienol A and difengpienol B, and two known compounds, dunnianol and isodunnianol, which all could inhibit inflammatory meditator NO production in LPS-stimulated RAW264.7 cells [8]. Simonsinol (Figure 1) is a sesqui-neolignan with similar structure to dunnianol and isodunnianol, which was firstly isolated from the bark of *Illicium simonsii* by Kouno in 1994 [9]. However, the biological activity of simonsinol has not been reported so far. Thus, the anti-inflammatory activity and mechanism of simonsinol was investigated with an LPS-stimulated murine macrophage RAW264.7 cells model in this study.

## 2. Results

### 2.1. Effect of Simonsinol on RAW264.7 Cells Viability

In order to assess the cytotoxic effect of simonsinol on RAW264.7 cells, an MTT assay was carried out. Four concentrations of simonsinol, including 5, 10, 20, and 40 μM, were chosen to treat RAW264.7 cells for 24 h. As shown in Figure 2, compared to control cells’ treatment with 0.1% DMSO, the cell viabilities of the following treatments with 5, 10, 20, and 40 μM simonsinol were 99.4%, 98.2%, 97.6%, and 44.6%, respectively, which indicated that RAW264.7 cells viability was not affected significantly by up to 20 μM simonsinol for 24 h. Therefore, the concentration range of simonsinol for subsequent anti-inflammatory activity screening was determined to be within 20 μM.

### 2.2. Effect of Simonsinol on LPS-Stimulated RAW264.7 Cells Morphology

In order to test the anti-inflammatory activity of simonsinol, an LPS-stimulated RAW264.7 macrophages inflammatory model was set up. As shown in Figure 3, compared with the blank control cells (Figure 3a), after stimulation by LPS for 24 h, most of the RAW264.7 macrophages formed pseudopodia and showed larger size (Figure 3b). The significant morphological changes demonstrated that the LPS-stimulated inflammatory model was successful. While being treated with simonsinol for 1 h before LPS stimulation, the degree of morphological changes reduced dose dependently (Figure 3c–f). Especially, while being treated with 20 μM simonsinol, the cell morphology and size were similar to the control cells, indicating simonsinol could inhibit inflammatory response in LPS-stimulated RAW264.7 cells.

### 2.3. Simonsinol Inhibits NO, TNF-α, and IL-6 Production in LPS-Stimulated RAW264.7 Cells

NO, TNF-α, and IL-6 are important inflammatory meditators that are produced during the inflammation response process. To confirm the anti-inflammatory activity of simonsinol on LPS-stimulated RAW264.7 cells, the released NO was determined with Griess reagents, and the released TNF-α and IL-6 were determined with respective ELISA kits. Compared with the blank group cells, owing to LPS stimulation, NO, TNF-α, and IL-6 secretion in model group cells markedly increased. During treatment with 5, 10, and 20 μM simonsinol, the NO, TNF-α, and IL-6 concentration reduced dose dependently (Figure 4a–c), confirming that simonsinol could inhibit inflammatory response in LPS-stimulated RAW264.7 cells.

### 2.4. Simonsinol Suppresses iNOS, TNF-α, and IL-6 Transcription in LPS-Stimulated RAW264.7 Cells

To clarify the reason for the anti-inflammatory effect of simonsinol, transcription levels of iNOS, TNF-α, and IL-6 in LPS-stimulated RAW264.7 cells were determined. As shown in Figure 5a–d, irritation with LPS markedly upregulated the transcription level of iNOS, TNF-α, and IL-6 mRNA. Treatment with simonsinol could antagonize the effect of LPS, and dose dependently suppress the transcription level of iNOS, TNF-α, and IL-6 mRNA, demonstrating the anti-inflammatory effect of simonsinol via downregulating the transcription of inflammatory responsive genes such as iNOS, TNF-α, and IL-6.

### 2.5. Simonsinol Suppresses IκBα Phosphorylation in LPS-Stimulated RAW264.7 Cells

Activation of NF-κB is critically required for the expression of iNOS, TNF-α, and other inflammation responsive genes [10]. The pivot for activation of NF-κB is to phosphorylate IκBs. Once IκBs have been phosphorylated, they are ubiquitinated and degraded by 26S proteasome, thus, releasing a free NF-κB dimer which is translocated to the nucleus and induces the transcription of target genes [11]. Therefore, the phosphorylation of IκBα (p-IκBα) was examined. As shown in Figure 6, after induction with LPS for 1 h, the protein level of p-IκBα was upregulated remarkably, suggesting that NF-κB was activated by LPS. While treatment with simonsinol could antagonize the effect of LPS and dose dependently block the phosphorylation of IκBα, this demonstrates simonsinol could suppress inflammation responsive genes transcription through inactivation of the NF-κB signaling pathway.

## 3. Discussion and Conclusions

Macrophages play an essential role in inflammation response and they have three important functions: antigen presentation, phagocytosis, and immunomodulation which is through the production of a series of inflammatory meditators [2]. Thus, in this study, an LPS-stimulated RAW264.7 macrophages inflammatory model was adopted; this model also was used in many previous anti-inflammatory studies [12]. The data showed that 100 ng/mL LPS could induce significant morphological changes in RAW264.7 cells (Figure 3a,b), and increase the production of inflammatory meditators such as NO, TNF-α, and IL-6 (Figure 4), demonstrating that the LPS-stimulated RAW264.7 cells inflammatory model was set up successfully.

Pseudopodia formation is the characteristic morphological change of macrophages in inflammation response [13,14]. Simonsinol could antagonize the effect of LPS on the morphology of RAW264.7 cells, inhibiting the pseudopodia formation of RAW264.7 cells (Figure 3b–f). NO, TNF-α, and IL-6 are three important inflammatory meditators. NO, a free gaseous signaling molecule, takes part in the regulation of the nervous, cardiovascular, and immune system [15]. Simonsinol could effectively inhibit NO production in LPS-stimulated RAW264.7 cells (Figure 4a). TNF-α plays an important role in immunomodulation and is involved in Alzheimer’s disease and psoriasis [16,17]. IL-6 could promote the differentiation of lymphocytes and the release of other inflammatory cytokines, and also, could activate several important pathways in tumorigenesis [18,19,20]. TNF-α and IL-6 secretion in LPS-stimulated RAW264.7 cells were inhibited by simonsinol concentration dependently (Figure 4b,c). These data strongly demonstrate that simonsinol could relieve inflammation response in LPS-stimulated RAW264.7 cells.

The NF-κB signaling pathway is critically the control center for the regulation of inflammation [10]. Due to binding with IκBs, NF-κB normally is sequestered in the cytosol as an inactive complex in unstimulated cells. Once IκBs have been degraded, a free NF-κB dimer translocates to the nucleus and initiates the transcription of target genes [11]. In LPS-stimulated RAW264.7 cells, simonsinol could inhibit the phosphorylation of IκBα dose dependently (Figure 6), suggesting simonsinol could block the activation of the NF-κB signaling pathway. This was in accord with the result of RT-PCR, as the transcriptions of iNOS, TNF-α, and IL-6 mRNA, which under the regulation of NF-κB, were inhibited by simonsinol, too (Figure 5). The phosphorylation of IκBα was under the control of cascaded kinases, including IκB kinases (IKKs) and their upstream NF-κB-inducing kinase (NIK) [21,22], simonsinol maybe could inhibit IKK or NIK, thus, inhibiting the phosphorylation of IκBα. However, which kinase was the direct target of simonsinol and whether simonsinol could regulate other signaling pathways need further investigation.

The *Illicium* plants are well-known as traditional folk medicines used to treat rheumatic arthritis and relieve pain in China. The anti-inflammatory activity of sesquiterpenes, neolignans, phenylpropanoids, phytoquinoids, and essential oils in *Illicium* plants has been reported before [23,24,25]. Until 2019, the anti-inflammatory activities of sesqui-neolignans, a type of natural product that exists widely in *Illicium* plants, were reported [8]. The data in this study further showed that sesqui-neolignans are another class of secondary metabolites with potential anti-inflammatory activity in *Illicium* plants.

In conclusion, the presented data in this study demonstrate that simonsinol could inhibit inflammation response in LPS-stimulated RAW264.7 cells through the inactivation of the NF-κB signaling pathway, indicating that it may be a potential anti-inflammatory agent.

## 4. Materials and Methods

### 4.1. General Information

NMR spectra were measured on an AVANCE III HD 500 spectrometer (Bruker Co., Ltd., Ettlingen, Germany). HR-ESI-MS data were obtained on a LC/MS-IT-TOF mass spectrometer (Shimadzu Co., Ltd., Kyoto, Japan). Compound separation was performed on silica gel (100–200 mesh, Qingdao Marine Chemical Factory, Qingdao, China) column chromatography.

### 4.2. Reagents

LPS and 3-(4,5-Dimethylthiazol-2-yl)-2,5-diphenyltetrazolium bromide (MTT) were purchased from Sigma-Aldrich (St. Louis, MO, USA). The Griess reagents, cell lysis buffer for Western blot, protease inhibitor cocktail, BCA protein assay kit, and enhanced chemiluminescence (ECL) kit were obtained from the Institute of Beyotime Biotechnology (Haimen, China). The ELISA kits for TNF-α and IL-6 were purchased from Elabscience Biotechnology Co., Ltd. (Wuhan, China). The RNAfast200 Extraction Kit used for total RNA extraction was purchased from Fastagen Biotechnology (Shanghai, China). The HiFi-MMLV cDNA Kit used for cDNA synthesis was obtained from CoWin Biosciences (Beijing, China). Mouse monoclonal antibody against β-tubulin was acquired from Proteintech Group (Wuhan, China). Rabbit monoclonal antibody against p-IκBα, horseradish peroxidase (HRP)-labeled goat anti-mouse IgG, and HRP-labeled goat anti-rabbit IgG were obtained from the Institute of Beyotime Biotechnology (Haimen, China).

### 4.3. Plant Material

*Illicium simonsii* were collected from Xuanwei City, Yunnan Province, China, in 2019 and identified by Prof. Zheng-hong Pan (Guangxi Institute of Botany, Chinese Academy of Sciences). A voucher specimen (CTM201910) was deposited at the Guangxi Key Laboratory of Functional Phytochemicals Research and Utilization, Guangxi Institute of Botany, Chinese Academy of Sciences, China.

### 4.4. Extraction and Isolation

The powdered bark of *Illicium simonsii* (10 kg) was extracted three times with dichloromethane (3 × 30 L) at room temperature. The solution was concentrated under reduced pressure to yield a crude extract (382 g), which was then separated on a silica gel column eluted with petroleum ether/acetone (50/1 to 5/1, gradient system) to give thirteen fractions (Frs. 1–13). Fr. 10 (5.4 g) was further purified by repeated silica gel column chromatography (petroleum ether/acetone, 7/1) to yield compound **1** (92.4 mg). Compound **1** possesses a molecular formula of C_27_H_26_O_3_ on the basis of HR-ESI-MS at *m*/*z* 397.1778 [M – H]^−^ (calculated 397.1809), and its ^1^H and ^13^C NMR data were consistent with those of simonsinol [9], which was also supported by 2D NMR, and the purity of compound **1** was more than 95% as confirmed by HPLC, see the Appendix A.

### 4.5. Cell Culture

The RAW264.7 cell line was acquired from the cell bank of the Chinese Academy of Sciences (Shanghai, China), and cultured in Dulbecco’s Modified Eagle’s Medium (DMEM) supplemented with 10% (*v*/*v*) heat-inactivated FBS at 37 °C in a humidified atmosphere containing 5% CO_2_. The cells were subcultured when they grew to 80–90% confluence and the culture media were replaced every other day.

### 4.6. MTT Assay

Single RAW264.7 cells suspensions with a density of 10^5^/mL (100 μL/well) were plated to a 96-well flat-bottom microtiter plate and cultured overnight. Then, the cells were treated with 5, 10, 20, or 40 μM simonsinol. The control cells were treated with equivalent DMSO. After 24 h, 10 μL MTT stock (5 mg/mL) was added to each well and incubated for 2 h. Then, the supernatants were discarded and 100 μL DMSO was added to each well. After 10 minutes of shaking, the optical density at the wavelength of 570 nm (OD570) was measured on a SPARK microplate reader (Tecan, Männedorf, Switzerland). All samples were tested in triplicate and repeated 3 times.

### 4.7. Effect of Simonsinol on LPS-Stimulated RAW264.7 Cells Morphology

Single RAW264.7 cells suspensions with a density of 10^5^/mL (2 mL/well) were seeded to a 6-well flat-bottom microtiter plate and cultured overnight. Then, the cells were pretreated with 5, 10, and 20 μM simonsinol or equivalent DMSO for 1 h. Subsequently, the cells were treated with or without 100 ng/mL LPS for 24 h and observed under an inverted microscope (Leica DMi1, Wetzlar, Germany).

### 4.8. Determination of Inflammatory Mediators in LPS-Induced RAW264.7 Cells

Single RAW264.7 cells suspensions with a density of 10^5^/mL (2 mL/well) were seeded to a 6-well flat-bottom microtiter plate and cultured overnight. Then, the cells were treated with 5, 10, and 20 μM simonsinol or equivalent DMSO for 1 h. Subsequently, the cells were treated with or without 100 ng/mL LPS for 24 h. After that, the cell culture media were collected, assayed for NO with Griess reagents, or assayed for TNF-α and IL-6 with respective ELISA kits according to the manufacturer’s instructions.

### 4.9. RT-PCR

RAW264.7 cells were pretreated with 5, 10, and 20 μM simonsinol or equivalent DMSO for 1 h and then, stimulated with or without 100 ng/mL LPS for 24 h. After that, total RNA from the cells was extracted using the RNAfast200 Extraction Kit. First strand cDNA was generated from 1 μg of total RNA with the HiFi-MMLV cDNA Kit. The primers used for amplification were as follows:

GAPDH forward and reverse primers, 5′-CTTTGTCAAGCTCATTTCCTGG-3′ and 5′-TCTTGCTCAGTGTCCTTGC-3′; iNOS forward and reverse primers, 5′-TTTGACGCTCGGAACTGTAG-3′ and 5′-GAGCCTGAAGTCATGTTTGC-3′; TNF-α forward and reverse primers, 5′-CTTCTGTCTACTGAACTTCGGG-3′ and 5′-CAGGCTTGTCACTCGAATTTTG-3′; IL-6 forward and reverse primers, 5′-GAGACTTCACAGAGGATACCAC-3′ and 5′-TCAGAATTGCCATTGCACAAC-3′.

PCR amplifications for 25 cycles were carried out under the following conditions: denaturation at 94 °C for 30 s, annealing at 55 °C for 30 s, and extension at 72 °C for 30 s. Amplification products were electrophoresed on a 1.5% agarose gel and visualized by staining with Gel Red.

### 4.10. Western Blot

RAW264.7 cells were treated with 5, 10, and 20 μM simonsinol or equivalent DMSO for 1 h, then, stimulated with or without 100 ng/mL LPS for another 1 h. Total protein from RAW264.7 cells was extracted with lysis buffer for Western blot adding protease inhibitor cocktail on ice and quantified by BCA protein assay kit. Proteins (20 μg) were separated by 10% SDS-PAGE and transferred to nitrocellulose membranes (0.45 μm). Nitrocellulose membranes were blocked with 5% (*w*/*v*) skimmed milk powder in Tris-buffered saline Tween 20 (TBST, 10 mM Tris, 150 mM NaCl, pH 7.4, 0.5% Tween 20) at room temperature for 2 h, and then, washed with TBST three times. Subsequently, the membranes were incubated with specific primary antibodies (anti-β-tubulin, anti-p-IκBα) at 4 °C overnight. After washing three times with TBST, the membranes were incubated with HRP-labeled secondary antibodies at room temperature for 2 h and then, washed three times with TBST again. Finally, the antibody-reactive bands were visualized with the ECL kit.

### 4.11. Statistical Analysis

All data used are expressed as means ± SD. Image Lab 3.0 (Hercules, CA, U.S.A.) was used to analyze the bands intensities of PCR and Western blot. Statistical analysis was performed with GraphPad Prism 5 software (La Jolla, CA, U.S.A.). Statistical differences were analyzed by unpaired *t*-test, *p* values < 0.05 were denoted by *, and *p* values < 0.01 were denoted by ** or ^++^.

## Figures and Tables

**Figure 1 molecules-25-03573-f001:**
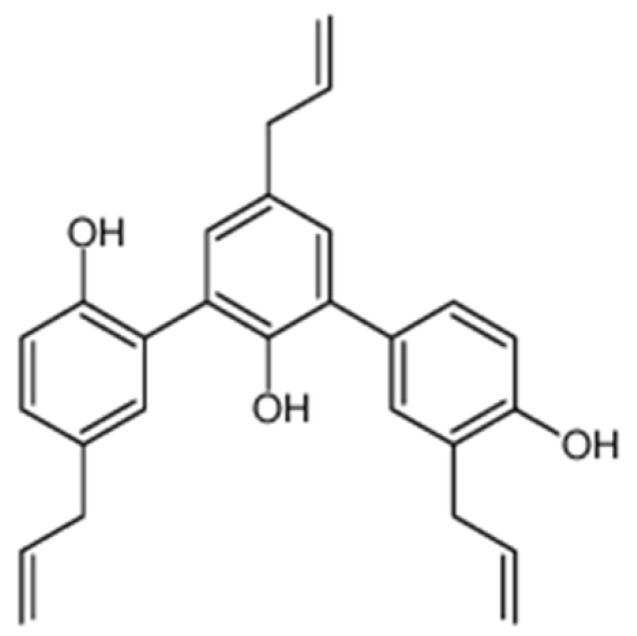
Structure of simonsinol.

**Figure 2 molecules-25-03573-f002:**
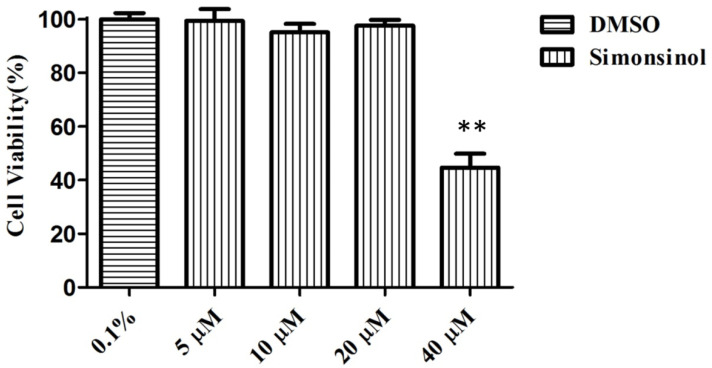
Effects of different concentrations of simonsinol on RAW264.7 cells viability. The cells viability (%) = (ODsimonsinol − OD_DMSO_)/OD_DMSO_ × 100. **, highly significant difference compared to control cells that treated by DMSO. (*p* < 0.01).

**Figure 3 molecules-25-03573-f003:**
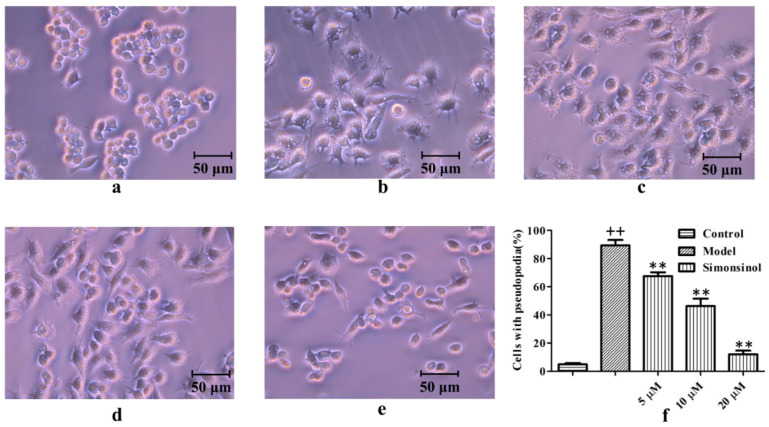
Effect of simonsinol on LPS-stimulated RAW264.7 cells morphology. (**a**) Blank control RAW264.7 cells. (**b**) Model RAW264.7 cells stimulated by LPS for 24 h. (**c**–**e**) RAW264.7 cell treated by 5, 10, and 20 μM simonsinol for 1 h, then, stimulated by LPS for 24 h. (**f**) Percentage of cells that formed pseudopodia. ^++^, highly significant difference compared to blank control cells (*p* < 0.01). **, highly significant difference compared to LPS-stimulated model cells (*p* < 0.01).

**Figure 4 molecules-25-03573-f004:**
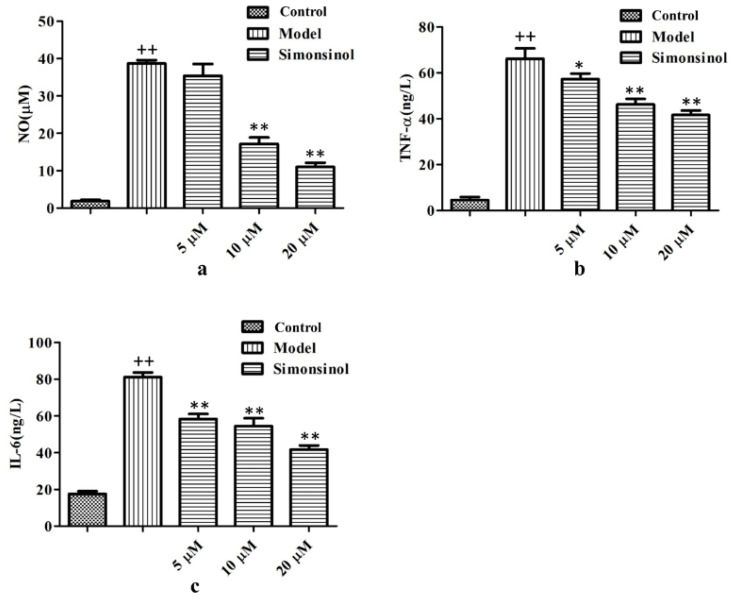
Simonsinol inhibits NO (**a**), TNF-α (**b**), and IL-6 (**c**) secretion in LPS-stimulated RAW264.7 cells. Data are represented as the mean ± SD (*n* = 3). ^++^, highly significant difference compared to blank control cells (*p* < 0.01). *, significant difference compared to LPS-stimulated model cells (*p* < 0.05), **, highly significant difference compared to LPS-stimulated model cells (*p* < 0.01).

**Figure 5 molecules-25-03573-f005:**
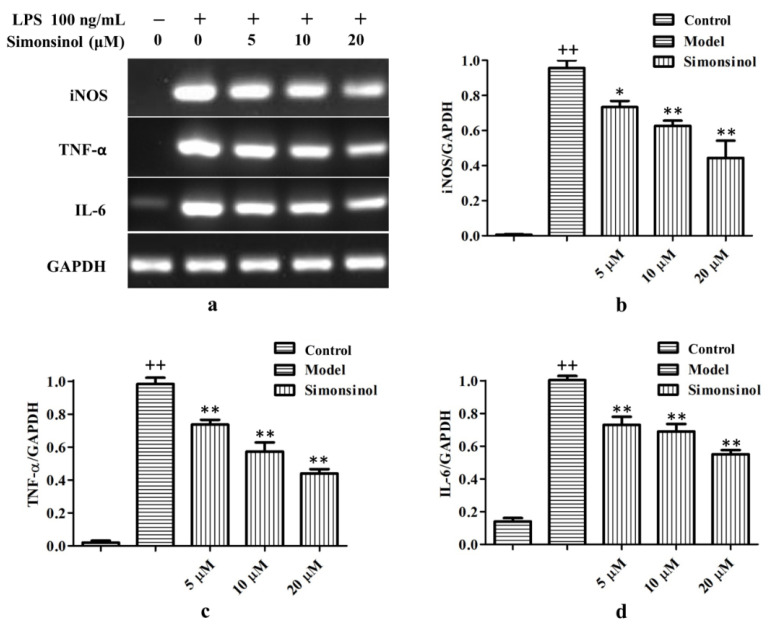
Simonsinol dose dependently suppresses transcription of iNOS, TNF-α, and IL-6 mRNA in LPS-stimulated RAW264.7 cells. (**a**) Representative result of PCR. (**b**) Relative expression of iNOS mRNA. (**c**) Relative expression of TNF-α mRNA. (**d**) Relative expression of IL-6 mRNA. Data are represented as the mean ± SD (*n* = 3). ^++^, highly significant difference compared to blank control cells (*p* < 0.01). *, significant difference compared to LPS-stimulated model cells (*p* < 0.05). **, highly significant difference compared to LPS-stimulated model cells (*p* < 0.01).

**Figure 6 molecules-25-03573-f006:**
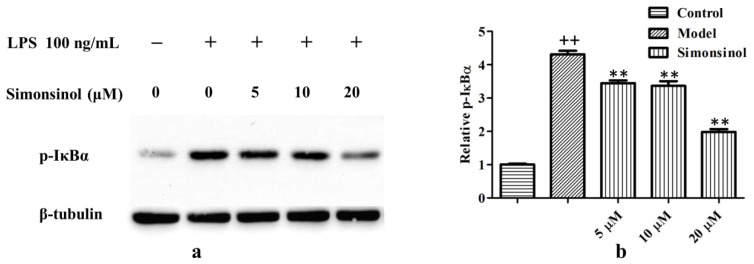
Simonsinol dose dependently suppresses phosphorylation of IκBα in LPS-induced RAW264.7 cells. (**a**) Representative result of Western blot. (**b**) Relative expression of p-IκBα. Relative p-IκBα = (sample intensity of p-IκBα/sample intensity of p-IκBα)/(blank intensity of p-IκBα/blank intensity of p-IκBα). Data are represented as the mean ± SD (*n* = 3). ^++^, highly significant difference compared to blank control cells (*p* < 0.01). **, highly significant difference compared to LPS-stimulated model cells (*p* < 0.01).

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
