# Peer review of "Anti-Inflammatory Effect of Simonsinol on Lipopolysaccharide Stimulated RAW264.7 Cells through Inactivation of NF-κB Signaling Pathway"

_molecules, 2020, doi:10.3390/molecules25163573_

Round 1

Reviewer 1 Report

This sentence: "Pseudopodia formation is the characteristic morphological change of macrophages in inflammation response" needs a reference.

The references 9, 22 and 23 did not report the anti-inflammatory activity of the sesqui-neolignans, as it seems to be described by the authors in the following sentence:

Until 2019, the anti-inflammatory activities of sesqui-neolignans, a type of natural products exist widely in Illicium plants, were reported [8, 9, 22, 23]. The data in this study further showed that sesqui-neolignans are another class of secondary metabolites with potential anti-inflammatory activity in Illicium plants.

In Material and Methods, what is TBST?

Do the authors consider that an assay with a synthetic anti-inflammatory (e.g. dexamethasone) could provide a supplemental information about the strength of the anti-inflammatory activity of the natural product when compared to a reference?

Author Response

For reviewer 1:

Thank you very much for your comments.

  1. This sentence: "Pseudopodia formation is the characteristic morphological change of macrophages in inflammation response" needs a reference.

  Response: Thank you for your suggestion. Two related references that also described pseudopodia formation of macrophages in inflammation response were added in the revised manuscript.

  1. The references 9, 22 and 23 did not report the anti-inflammatory activity of the sesqui-neolignans, as it seems to be described by the authors in the following sentence: 

Until 2019, the anti-inflammatory activities of sesqui-neolignans, a type of natural products exist widely in Illicium plants, were reported [8, 9, 22, 23]. The data in this study further showed that sesqui-neolignans are another class of secondary metabolites with potential anti-inflammatory activity in Illicium plants.

  Response: Sorry for our mistake, the references 9, 22 and 23 have been deleted in the revised manuscript.

  1. In Material and Methods, what is TBST?

 Response: Sorry for our neglect. In fact, TBST is abbreviation for Tris-buffered saline Tween 20 (10 mM Tris, pH7.4, 150 mM NaCl, 0.5% Tween 20), and in the revised manuscript the full name was given before TBST occurred at the first time.

  1. Do the authors consider that an assay with a synthetic anti-inflammatory (e.g. dexamethasone) could provide a supplemental information about the strength of the anti-inflammatory activity of the natural product when compared to a reference?

 Response: Yes, we agree that in comparison to a reference could provide supplemental information about the strength of the anti-inflammatory activity of the compound. In fact, during our experiments process, we chose a natural product parthenolide as reference. And parthenolide showed more powerful anti-inflammatory activity than simonsinol, with the IC50 for NO, TNF-alpha and IL-6 release were 0.27, 0.41 and 0.57 μM, respectively.

Reviewer 2 Report

The work by Li et al. showed that simonsinol suppresses the production of nitric oxide, TNF-alpha, and IL-6 by downregulating the NF-κB signaling pathway. Their manuscript is concise and well organized. The topic looks interesting. This reviewer thinks that the manuscript is worthy of publication; however, there are a couple of concerns.

Specific comments

Show the purity of simonsinol.

Discuss how simonsinol inhibits phosphorylation of IκBα.

There are several grammatical and editorial errors throughout the manuscript. The authors should make sure that there is no grammatical or editorial error in the manuscript before submission.
